# Heat flows solubilize apatite to boost phosphate availability for prebiotic chemistry

Thomas Matreux [1,7,8], Almuth Schmid [1,8], Mechthild Rappold[1], Daniel Weller [2], Ayşe Zeynep Çalışkanoğlu [2], Kelsey R. Moore[3,4], Tanja Bosak [4], Donald B. Dingwell [2], Konstantin Karaghiosoff[5], François Guyot[6], Bettina Scheu[2], Dieter Braun [1] & Christof B. Mast [1]✉

Phosphorus is an essential building block of life, likely since its beginning. Despite this importance for prebiotic chemistry, phosphorus was scarce in Earth's rock record and mainly bound in poorly soluble minerals, with the calcium-phosphate mineral apatite as key example. While specific chemical boundary conditions have been considered to address this so-called phosphate problem, a fundamental process that solubilizes and enriches phosphate from geological sources remains elusive. Here, we show that ubiquitous heat flows through rock cracks can liberate phosphate from apatite by the selective removal of calcium. Phosphate's strong thermophoresis not only achieves its 100-fold up-concentration in aqueous solution, but boosts its solubility by two orders of magnitude. We show that the heat-flow-solubilized phosphate can feed the synthesis of trimetaphosphate, increasing the conversion 260-fold compared to thermal equilibrium. Heat flows thus enhance solubility to unlock apatites as phosphate source for prebiotic chemistry, providing a key to early life's phosphate problem.

Phosphorus is an integral part of all life and is used in metabolites such as nucleoside-polyphosphates, phospholipids in cell membranes, and the backbone of DNA or RNA[1] (Fig. 1a). The abundance of phosphorus in biomolecules is likely to be no accident but, on the contrary, dictated by function. Oligonucleotides, for example, become negatively charged under neutral conditions through the phosphate backbone, which is essential to keep them readable and prevent hydrolysis[2]. Given this central importance, it is assumed that phosphate must have been available at an early stage during the origins of life.

However, in the presence of divalent ions[3], phosphate is water-insoluble over a wide pH range and, most importantly, precipitates with calcium to form apatite $Ca_5(PO_4)_3(F, Cl, OH)$ or brushite $CaHPO_4 \cdot 2 H_2O$. These processes likely reduced the amount of free phosphate 4 billion years ago, keeping it far below the high millimolar concentration regime required for prebiotic chemistry[4–8]. This high demand also stems from the poor reactivity of orthophosphate, e.g., for the phosphorylation of nucleosides[4–6] and precursors[7,8], efficient reaction buffering[4,8], and the synthesis of phosphate species suitable to drive key reactions of prebiotic chemistry[8–11]. In combination with its low availability, this constitutes the so-called phosphate problem, which has been identified as one of the major hurdles to understanding the origins of life on Earth[2,12].

[1]Systems Biophysics, Ludwig Maximilians University, Munich, Germany. [2]Earth and Environmental Sciences, Ludwig Maximilians University, Munich, Germany. [3]Department of Earth and Planetary Sciences, Johns Hopkins University, Baltimore, MD, USA. [4]Department of Earth, Atmospheric and Planetary Sciences, Massachusetts Institute of Technology, Cambridge, MA, USA. [5]Department of Chemistry, Ludwig Maximilians University, Munich, Germany. [6]Institut de Minéralogie, de Physique des Matériaux et de Cosmochimie (IMPMC), MNHN, CNRS, IRD, Sorbonne Université, Paris, France. [7]Present address: Laboratoire de Biophysique et Evolution, UMR CNRS-ESPCI 8231 Chimie Biologie Innovation, PSL University, Paris, France. [8]These authors contributed equally: Thomas Matreux, Almuth Schmid. ✉e-mail: christof.mast@physik.uni-muenchen.de

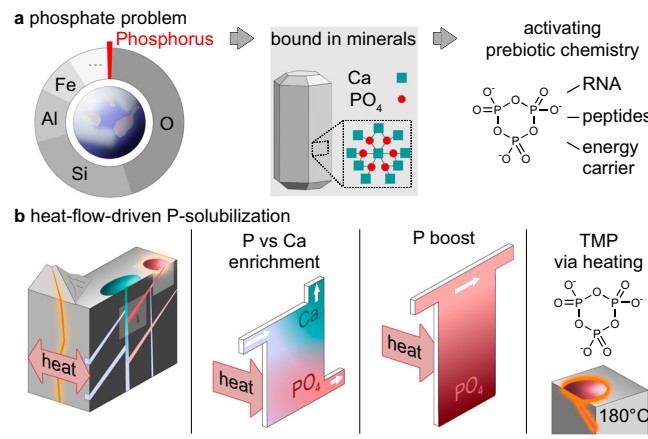

**Fig. 1 | A geothermal solution to the phosphate problem on the early Earth. a** Phosphorus only constitutes around 0.1 wt% of Earth's crust and is mostly bound as phosphate in apatite minerals, which renders it inaccessible for nascent life. **b** Heat flows in geothermal systems are able to enrich phosphate against calcium, boosting phosphate solubility at neutral pH and its absolute concentrations for downstream synthesis of energy-rich trimetaphosphate (TMP).

Apatite was presumably abundant compared to other phosphate-bearing minerals[12–16], for instance in frequent inclusions in igneous, sedimentary, and metamorphic rocks[15,16]. Therefore, strategies to release its bound phosphate for use in prebiotic chemistry have been investigated. While apatite easily dissolves under low pH conditions (1.5–2.5)[13,17], as found in lacustrine environments, acidic lakes[18,19], hot springs[20], and hydrothermal systems[21–24] with steep pH gradients[25–28], such environments are often incompatible with prebiotic chemistry, which commonly requires neutral to alkaline conditions[18,24,28]. Here, the dissolved phosphate quickly precipitates with presumably abundant calcium, so that despite the broad range of pH conditions available in geological systems, one is back at square one of the phosphate problem.

Chemically, this dilemma has been addressed with the help of chelating agents such as ammonium oxalate[29] or citric acid[30], which bind calcium and thus keep phosphate in solution even under neutral to alkaline conditions. At the same time, it raises the question of the plausibility of the high chelating agent concentrations required[31]. In carbonate lakes, sequestration of free calcium into calcium carbonate is possible, with acidic inflows constantly supplying new phosphate and evaporation increasing its concentration to 0.1 molal[32]. However, evaporation unselectively up-concentrates all ions, which is potentially detrimental to prebiotic chemistry reactions[33–35] and fatty acid vesicle[33] or coacervate stability[36].

The low reactivity of orthophosphate motivated the search for more reactive and soluble phosphorus-containing molecules. Reduced phosphorus of meteoritic origin has been shown to phosphorylate glycerol[37] and yield diamidophosphate[38]. Condensed phosphates such as cyclic trimetaphosphate (TMP)[39–42] that, however, usually require high temperatures as found in volcanic environments[43] (>500 °C) for their efficient synthesis have been shown to trigger peptide synthesis[41,44,45] or the phosphorylation of nucleosides[41,46]. Chemically activated phosphates such as phosphoenolpyruvate[8] and acetyl[10], carbamoyl[11], and imidazole phosphate[9] have been demonstrated to drive efficient phosphorylation of a variety of prebiotically relevant species, but also require high phosphate concentrations for their synthesis.

Thus, it becomes apparent that prebiotic chemistry would benefit massively from a chelating-independent and widely accessible process that drives the large-scale release of orthophosphate from geomaterials, protects it from diffusive dilution and enables its downstream condensation to polyphosphates. A promising scenario are permeable pathways in rocks, such as water-filled fractures in magmatic or

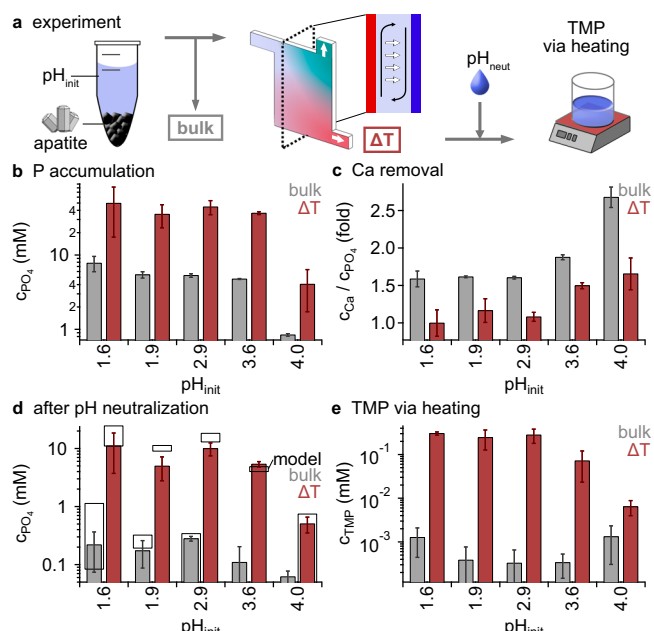

**Fig. 2 | Heat-flow-driven solubilization of phosphate from apatite. a** Experiment. Acidic-dissolved phosphate (pH 1.6 to pH 4), is flushed through a heat flow chamber (ΔT), leading to the selective enrichment of phosphate at the bottom outlet by the interplay of convection (black) and thermophoresis (white). Downstream neutralization (pH$_{neut}$) mimics the transition to prebiotic chemistry conditions that allow, for instance, the formation of trimetaphosphate (TMP). **b** Dissolved phosphate was accumulated from initially low concentrations (bulk, gray) and extracted in the bottom outlet of the heat flow chamber (ΔT, red). **c** Here, calcium concentrations were depleted relative to phosphate by the thermal non-equilibrium, which shifts the Ca:PO$_4$ ratio from 5:3 found in apatite (bulk, gray) to 1:1 (ΔT, red) (see Supplementary Fig. 3). **d** Under the neutral conditions required for prebiotic chemistry, previously acidic-dissolved phosphate and calcium precipitated (see Supplementary Figs. 4, 5). In contrast to the bulk case (gray), the heat-flow-driven removal of calcium (ΔT, red), boosted the solubility of phosphate up to 100-fold. The results were verified by geochemical modeling (black boxes, indicating mean ± SD, see "Methods"). **e** Moderate heating to 180 °C triggered the formation of TMP from the heat-flow-altered solutions (ΔT, red), increasing yields more than 100-fold compared to the absence of thermal gradients (bulk, gray). All error bars show the SD.

geothermal environments, as well as sedimentary layers of shallow submarine or lacustrine settings that are exposed to heat fluxes. Such thermo-microfluidic environments have been investigated as a simple but ubiquitous and versatile tool that drives molecular selection and prebiotic reactions[45]. Here, the superposition of thermally driven thermophoresis and convection could have enriched prebiotic building blocks depending on their charge, size and type, ultimately boosting downstream prebiotic chemistry through the shift of solute compositions towards reactive species[35,45,47].

Heat sources are widespread as a universal consequence of the second law of thermodynamics[48], e.g., found near meteoritic impacts[49], in volcanic environments[50], or in hydrothermal systems[51]. This raises the question whether heat flows could offer a new pathway for broad prebiotic availability of phosphate from minerals and other geomaterials.

Here, we show that heat flows through thin rock fractures can selectively shift the composition of solutions from phosphate-rich minerals such as apatite and thus prevent their precipitation under the close-to-surface neutral pH conditions relevant for prebiotic chemistry. Molecule-selective thermogravitational enrichment, thus, has a similar effect as chemical approaches that actively remove the calcium responsible for phosphate precipitation (Fig. 2). While apatite

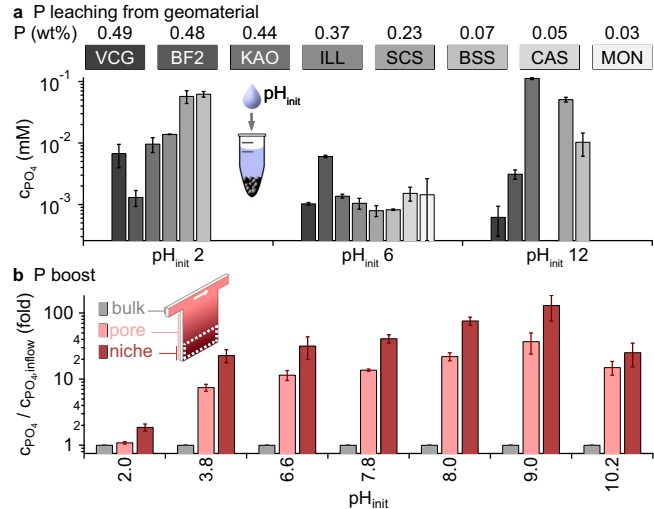

**Fig. 3 | Phosphate-rich habitats formed by heat-flow-driven accumulation.**
**a** Leaching from geomaterials with different weight percentages of phosphorus (see Supplementary Table 2) in solutions of different $pH_{init}$ yielded low phosphate concentrations: VCG volcanic glass, BF2 basalt F2, KAO kaolinite, ILL illite, SCS siliciclastic sand, BSS basalt sand, CAS carbonate sand, MON montmorillonite (for SEM images and compositions see Supplementary Figs. 7, 8 and Supplementary Table 2). **b** Heat flows across water-filled fractures boosted the phosphate concentrations from the top-feeding inflows ("bulk", gray) by a factor of 130-fold in the lowest 25 % of the pore ("niche", dark red), or 40-fold averaged over the whole crack ("pore", light red). All error bars show the SD.

dissolves in acidic flows, the resulting solution would re-precipitate in shallow, neutral waters under thermal equilibrium conditions. We show that heat-flow-driven enrichment in acidic solutions can supply up to 15 millimolar of solubilized phosphate after close-to-surface pH neutralization, leading to TMP formation under subsequent moderate heating. We extend this concept to a wide range of further geomaterials, such as clays and sands, and show that the highly dilute phosphate in their leachates can be thermophoretically accumulated within porous niches more than 100-fold (Fig. 3). Our experimental findings are consistent with geochemical modeling[52,53] and demonstrate an active coupling of geological leaching and precipitation effects with thermal non-equilibrium systems. The combined process can thereby unlock a widely available phosphate source on the prebiotic Earth.

## Results

### Apatite solubilization under equilibrium conditions

Apatite and especially fluorapatite are assumed to be among the most abundant phosphate sources on the early Earth[12–14]. However, the solubility of phosphate from apatite strongly depends on pH, peaking at acidic conditions (Supplementary Fig. 1) that are widely incompatible with prebiotic chemistry[4,6–8,10], and is essentially insensitive to temperature, mass-to-volume ratio, and grain size (Supplementary Fig. 2)[17]. To obtain an experimental baseline without the effect of heat-flow-driven enrichment, we first characterized the amount of phosphate leached from three natural apatite samples under bulk conditions[17,35] (Ipirá Complex, Brazil, Durango, Mexico and Ontario, Canada, see Table 1 and Supplementary Fig. 1). We exposed them to a wide range of pH conditions between pH 1 and pH 12. After reaching an equilibrium in pH and solute concentrations (Supplementary Fig. 2), we found that the amount of leached phosphate was largely consistent across the different samples and in agreement with previous literature studies[17]. The results confirmed the characteristic strong dissolution at acidic pH (see Table 1) with up to 5 mM leached phosphate concentration at pH 1, but low phosphate concentrations of ~10 μM at more neutral pH values compatible with prebiotic chemistry. In the

## Table 1 | Compositions of used apatites

| Oxide composition | Brazil, Ipirá complex (%) | Canada, Ontario (%) | Mexico, Durango (%) |
|---|---|---|---|
| CaO (%) | 54.6 ± 0.7 | 55.0 ± 0.3 | 54.0 ± 0.4 |
| $P_2O_5$ (%) | 39.0 ± 0.7 | 41.5 ± 0.6 | 41.3 ± 0.7 |
| $SiO_2$ (%) | 1.45 ± 0.09 | 0.17 ± 0.03 | 0.27 ± 0.02 |
| $Na_2O$ (%) | 0.026 ± 0.012 | 0.19 ± 0.09 | 0.29 ± 0.02 |
| FeO (%) | 0.006 ± 0.009 | 0.018 ± 0.012 | 0.039 ± 0.013 |
| F (ppm) | 2.88 ± 0.79 | 2.42 ± 0.09 | 3.21 ± 0.93 |
| Cl (ppm) | 0.15 ± 0.06 | 0.003 ± 0.005 | 0.30 ± 0.07 |
| **Leaching conditions** | **$PO_4$ (mM)** | **$PO_4$ (mM)** | **$PO_4$ (mM)** |
| $pH_{init}$ = 2 | 2.49 ± 0.11 | 1.93 ± 0.03 | 2.09 ± 0.09 |
| $pH_{init}$ = 3 | 0.33 ± 0.02 | 0.28 ± 0.01 | 0.39 ± 0.01 |
| $pH_{init}$ = 6 – 7 | 0.004 ± 0.002 | 0.007 ± 0.003 | 0.007 ± 0.001 |

Measurements by XRF as described in Methods. Further results from leaching experiments over a wide pH range are shown in Supplementary Figs. 1, 2. The apatite Brazil sample is used in Fig. 2 and Table 2. All error values indicate the SD.

following experiments, we focused on the pH range from 1.6 to 4, as found in acidic geothermal flows[18–24]. More alkaline conditions would only trigger the marginal release of phosphate from apatite as shown in Supplementary Fig. 2a–c and visible in the drop of phosphate dissolution at pH 4 in Fig. 2b. Here, even heat-flow-induced fractionation of calcium and phosphate would not allow high final concentrations of phosphate.

In contrast to such closed laboratory systems, the open geological systems we consider here would maintain a constant pH at an externally-defined value through coupling to acidic geothermal flows[18–24]. Although a larger quantity of dissolved phosphate would be expected in such open systems, precipitation with calcium would lead to vanishing phosphate concentrations in solution under the more neutral pH conditions close to the surface that are required for pre-biotic chemistry[18,24,28].

To study this scenario experimentally (Fig. 2a), we prepared 5 samples in which ~3 g of apatite (Brazil apatite, see Table 1) was dissolved by repeatedly supplying HCl to reach the respective predefined initial pH values (shown in Fig. 2b–e). This procedure resulted in the apatite-specific calcium-to-phosphate ratios of 5:3. Given that a certain degree of dilution is expected in an open system due to the geothermal flows on its way to the surface, we have approximated this effect by diluting the above samples 4-fold (see "Methods"), yielding absolute concentrations of 7.8 mM of phosphate and 12.2 mM of calcium (Fig. 2b, c gray bars and Table 2). We mimicked the transition to neutral, near-surface pH conditions by repeated addition of NaOH until a constant neutral pH was reached (see Fig. 2d, Supplementary Fig. 4 and Methods, Eqs. 1–3). As expected, the resulting phosphate concentrations were at micromolar levels due to its nearly-complete precipitation with the calcium ions present (Fig. 2d, gray bar). The precipitation equilibria (Fig. 2d, black hollow boxes) were also modeled using PHREEQC-software[52,53], yielding a good agreement with the experimental findings. Analysis of precipitates by SEM and geochemical modeling[52,53] revealed the formation of calcium phosphates such as apatite (see Supplementary Table 1, Supplementary Figs. 4, 5). Different final pH conditions (8, 10, 12) and temperatures (30 °C, 60 °C, 90 °C) that may occur in a natural setting resulted in the same precipitation characteristics and phosphate concentrations (see Supplementary Fig. 4).

### Heat-flow-driven solubilization

The characteristics of this natural microfluidic system massively changed in the presence of heat flows that spatially separate

**Table 2 | Phosphate concentrations after dissolution, and pH neutralization and conversion to trimetaphosphate (TMP) in absence and presence of thermal non-equilibria**

| $pH_{init}$ | | dissolved $PO_4$ (mM) | after ΔT accumulation $PO_4$ (mM) | $PO_4$:Ca | after pH neutralization $PO_4$ (mM) | TMP (mM) | conversion (%) $PO_4 \rightarrow$ TMP |
|---|---|---|---|---|---|---|---|
| 1.6 | bulk | 7.8 ± 1.8 | | 1.6 ± 0.11 | 0.22 ± 0.15 | 0.0013 ± 0.0008 | 0.05 ± 0.02 |
| | ΔT | | 50 ± 32 | 1.0 ± 0.18 | 11.1 ± 7.4 | 0.30 ± 0.03 | 10.3 ± 0.8 |
| 1.9 | bulk | 5.4 ± 0.5 | | 1.6 ± 0.01 | 0.17 ± 0.09 | 0.0004 ± 0.0004 | 0.02 ± 0.02 |
| | ΔT | | 35 ± 12 | 1.2 ± 0.16 | 5.0 ± 2.2 | 0.25 ± 0.12 | 12.8 ± 6.1 |
| 2.9 | bulk | 5.3 ± 0.3 | | 1.6 ± 0.02 | 0.28 ± 0.03 | 0.0003 ± 0.0003 | 0.02 ± 0.02 |
| | ΔT | | 44 ± 10 | 1.1 ± 0.06 | 10.0 ± 2.5 | 0.28 ± 0.10 | 15.9 ± 5.1 |
| 3.6 | bulk | 4.8 ± 0.1 | | 1.9 ± 0.03 | 0.11 ± 0.09 | 0.0003 ± 0.0002 | 0.02 ± 0.01 |
| | ΔT | | 37 ± 2 | 1.50 ± 0.04 | 5.4 ± 0.6 | 0.07 ± 0.05 | 4.6 ± 3.1 |
| 4 | bulk | 1.5 ± 0.04 | | 2.7 ± 0.14 | 0.06 ± 0.02 | 0.0013 ± 0.0010 | 0.5 ± 0.4 |
| | ΔT | | 9 ± 3 | 1.65 ± 0.21 | 0.5 ± 0.2 | 0.006 ± 0.002 | 2.3 ± 0.8 |

All error values indicate the SD.

phosphate and calcium ions before reaching the more neutral environments near the surface, where unaltered apatite-leachates would precipitate completely. We experimentally implemented such a setting as plausible in heated rock cracks[49–51] using an open microfluidic heat flow chamber with a thickness of 200 μm across and applied a thermal gradient of 20 K (see Fig. 2a, Supplementary Fig. 6). As discussed below, more shallow thermal gradients in larger pore networks are expected to lead to the same results. In this flow-through system, we set the inflow volume rate[54] of acidic-dissolved apatite solution to 15 nl s⁻¹. The outflow volume rates were set to 0.75 nl s⁻¹ (bottom) and 14.25 nl s⁻¹ (top). The experiment was run for a period of 3 days until sufficient sample volume was obtained for analysis using ion chromatography and for subsequent phosphate polymerization (see Fig. 2e and Supplementary Fig. 3e–g). Such moderate flow rates were shown to not disturb the selective accumulation of ions through concurrent solvent convection (black arrows) and solute thermophoresis (white arrows)[27,35,45].

We explored the same pH conditions between pH 1.6 and pH 4 for apatite dissolution, as in the equilibrium case above, to study the respective characteristics of phosphate enrichment. While without thermal gradients the near-surface neutralization of the dissolved apatite samples leads to a re-precipitation of phosphate with calcium, we hypothesized that the heat-flow-driven fractionation of the two ions may result in high phosphate concentrations remaining in solution even after pH neutralization.

Our experiments showed indeed that phosphate and calcium were selectively enriched in the respective outlet channels. As depicted in Supplementary Fig. 3, in the bottom outflow, phosphate was accumulated up to 70 % stronger than calcium, reaching absolute concentrations of 50 mM, compared to 53 mM of calcium (see Table 2). Accordingly, the thermal non-equilibrium shifted the calcium-to-phosphate ratio from the apatite-defined 5:3 to 1:1 (see Fig. 2c). This local excess of the anionic species was balanced by a reduction of pH of up to one unit, providing charge neutrality (Supplementary Fig. 3).

We again mimicked the passage of the heat-flow-enriched phosphate solution to close-to-surface pH neutrality by successively adding NaOH until equilibrium was reached at pH 7–9. While the precipitation of phosphate and calcium still occurred, far too few calcium ions were present to bind all the dissolved phosphate and thereby remove it from solution. This led to a significantly higher final phosphate concentration of 15 mM compared to the case without heat-flow-induced phosphate enrichment with only 0.2 mM (Fig. 2d). The results were consistent with geochemical modeling[52,53] using measured concentrations of all participating ions obtained by ion chromatography (see "Methods").

Thus, heat-flow-driven ion enrichment yielded an about 100-fold increase in phosphate solubility, offering new opportunities towards the synthesis of activated phosphate species[8–11,26,39,41,42,46]. To test this, we heated the neutral phosphate solutions obtained before, which were previously thermally fractionated (Fig. 2c "ΔT") or unaltered by heat flows (Fig. 2c "bulk"), to 180 °C and measured the amount of TMP formed. In the unaltered solutions, our comparably moderate[55] but also prebiotically more widely available heating only resulted in 1 μM TMP concentrations (Fig. 2d "bulk"). In contrast, up to 0.3 mM TMP was formed in the thermally fractionated solutions, showing a 260-fold boost in concentrations (Fig. 2d "ΔT"). While without heat-flows, only <0.5 % of the initially acidic dissolved phosphate was converted to TMP due to its re-precipitation with calcium, the thermal non-equilibrium boosted the conversion to up to 16 % of the starting material (see "Methods", Eq. (4) and Table 2). As TMP has been shown to accumulate efficiently through thermogravitational accumulation due to its high Soret coefficient[45], the synthesized TMP could be further up-concentrated in similar settings and used downstream for TMP-driven prebiotic reactions such as the in-situ dimerization of glycine[44,45].

### Establishment of phosphate-rich geo-habitats
However, nature could not afford to rely purely on apatite as the sole source of phosphate, which is why we investigated how heat flows could utilize and concentrate phosphate in geothermal streams fed by a variety of igneous, sedimentary, and metamorphic rocks that contain lower amounts of phosphate[56].

We, therefore, studied the leaching from various basalts, sands, and clays (SEM images of geomaterials are shown in Supplementary Figs. 5, 6 and compositions in Supplementary Table 2). Without thermal gradients, the resulting phosphate concentrations barely exceeded 0.1 mM (Fig. 3a). Leaching from basalt sand (BSS) showed to be most efficient for acidic pH (pH 2); for neutral pH, basalt F2 (BF2) provided most phosphate, yielding 60 μM. These low orthophosphate concentrations are insufficient to drive common prebiotic reactions and are so far only known to be enhanced by drying at gas interfaces, which is, however, restricted to near-surface environments. We, therefore, aimed to explore how heat-flow-driven accumulation could boost phosphate concentrations in ubiquitous liquid-only settings.

For this, we explored the thermogravitational accumulation in small, protected pockets, that could provide reaction niches for prebiotic chemistry. We used the same experimental setup as above, but closed the bottom outlet, creating a water-filled pore connected to a continuous flow of liquid at the top. We speculated that the closed pore, which lacks a lower material outflow, would maximize the heat-flow-driven phosphate accumulation. For each experiment at a defined pH value, we applied a volume flow rate of 30 nl s⁻¹, fed by reservoirs of this pH, each with 500 μM dilute phosphate over the course of 1 week ("bulk", Fig. 3b). While these flow rates generated flow velocities of

about 160 $\mu$m s$^{-1}$ in the inlet channels, they only influenced the convection currents in the uppermost 4 mm of the heat-flow chamber, so that an efficient accumulation of the phosphate could take place (Supplementary Fig. 9b). Due to the high volumes required for these experiments, we used freshly dissolved NaH$_2$PO$_4$ as a representative soluble phosphate.

After the experiment, we froze the entire chamber and segmented the frozen content along the height of the chamber into four fractions of equal volume. IC measurements of all fractions revealed an up to 130-fold concentration increase of orthophosphate in the lowest quarter fraction with a height of 12.5 mm (Fig. 3b "niche", Supplementary Fig. 9), corresponding to an absolute phosphate concentration of 65 mM, with an up to 40-fold boost of phosphate averaged over the entire pore (Fig. 3b "pore"). The geomaterial that served as phosphate source in Fig. 3a could also support the formation of polyphosphates (see Supplementary Fig. 10) and enhance the formation of TMP up to 5.2-fold on kaolinite.

These results demonstrate that heat flows can boost orthophosphate concentrations in all-liquid environments without the need for dry-wet cycles or gas-water interfaces, making this process a geologically widely accessible method for prebiotic chemistry.

## Discussion

In this work, we have studied the heat-flow-driven solubilization of phosphate from natural apatite and other phosphate-bearing geomaterials, and its subsequent up-concentration to form geological, phosphate-rich habitats for prebiotic chemistry. To tackle this crucial question, we have taken into account the wide range of pH conditions found in such geological systems: While apatite could be dissolved without heat flows under acidic conditions, the more neutral pH conditions required for prebiotic chemistry would lead to close-to-complete precipitation, rendering the phosphate inaccessible for downstream reactions.

However, once the acidic-dissolved apatite solution was exposed to heat fluxes before reaching pH-neutral conditions, the selective thermophoretic removal of the co-dissolved calcium led to a 100-fold increase in phosphate solubility at neutral pH. We demonstrated that the achieved phosphate concentrations of up to 15 mM strongly enhanced the formation of TMP, an important activating and phosphorylation agent for driving various prebiotic reactions. The same enrichment mechanism boosted even the low phosphate concentrations that would be obtained from a range of phosphate-bearing geomaterials by up to 130-fold. As illustrated in Supplementary Fig. 9, these high phosphate concentrations reached after 1 week do not yet reflect the final steady state of the system, but will increase to 1000-fold after 100 days. Due to the robustness of such systems to geometrical[57] or temporal[45] perturbations, significantly higher concentrations could be achieved[58,59], limited only by the duration of the accumulation process and the ionic strength of the solvent[60].

While the temperature gradients used in the experiments were comparably high (20 K) to reduce experimental timescales[57], it is known that the effect of heat-flow-driven enrichment scales with the system size, ultimately balancing lower thermal gradients in a natural system[35,45]. In this work, the phosphate leaching from geomaterial and the phosphate enrichment driven by the heat flow were carried out sequentially to maximize reproducibility and to clarify the key mechanisms. In a natural system, the walls of heat-penetrated rock cracks or sedimented grains themselves could contain phosphate-bearing material, so that both processes could, in principle, take place simultaneously. The resulting more complex crack shapes do not hinder accumulation[57] and the comparatively fast leaching process of apatite at the low pH values considered here at rates[17] of 10$^{-8}$ mol m$^{-2}$ s$^{-1}$ would provide continuous feeding and drive further accumulation in downstream chambers on which we focused in our experimental model.

In addition, steep local temperature gradients can occur in narrow cracks near wider fractures that host rapid geothermal flows acting as a continuous heat source[54]. Simple heat flows in geothermal environments thus create a protected niche where high phosphate concentrations established from solubilized apatite and other geomaterials could drive prebiotic reactions[4-8], act as a buffer[4,8], and yield activated phosphate species[8-11].

## Methods

### Materials

NaH$_2$PO$_4$, Na$_2$HPO$_4$, Na$_3$P$_3$O$_9$, and Na$_5$P$_3$O$_{10}$ were purchased from Sigma Aldrich (USA), MSA (Methanesulfonic acid) and Na$_4$P$_2$O$_7$ from CarlRoth GmbH (Germany). For all experiments, ion chromatography water was used (Fisher Scientific, USA). For calibration and reference, Dionex Seven Anion Standard II, and Dionex Combined Six Cation Standard II from Thermo Fisher (USA) were used.

### Apatite sample preparation

The composition of apatite samples is shown in Table 1. The samples were crushed with a hammer to an average grain size of 1–2 cm and ground by a vibration mill to <500 $\mu$m mesh size. The different fractions were separated by hand sieving to grain sizes between 500 and 63 $\mu$m. The different apatite samples were characterized using XRF and SEM/EDX (for details, see below), as depicted in Table 1 and Supplementary Table 2. The H$_2$O$_2$-washing of samples for comparison with unwashed samples (see Supplementary Fig. 2d) was done by incubating samples in H$_2$O$_2$ (10 vol%) for 7 days to remove biological contaminations, rinsing them with purified water, and drying for 24 h at 200 °C in a furnace (similar to[35]).

### Major and trace element analyses of apatite samples

For the apatite samples, major and trace element analyses were carried out at University of Mainz (Germany) using a Malvern PANalytical Axios Fast X-ray fluorescence spectrometer (XRF) (Spectris Plc, UK). The major element analyses were carried out on fused glass discs, the trace element analyses were carried out on compacted powder pellets. Typical accuracy of the analyses of the standard references was ~1% relative (RMS) for major elements and 4% relative (RMS) for trace elements.

### Preparation of acidic apatite solution

Large stocks of Brazil apatite were mixed with IC grade water and adjusted multiple times with HCl to the desired pH. We waited for 2–3 weeks between adjustments until the pH was steady. Before the experiments, solutions were diluted with three parts of IC grade water (adjusted to experimental pH) and filtered (0.22 $\mu$m).

### pH measurement

The pH values were measured using a Thermo Scientific™ Orion™ 8220BNWP pH Electrode (Thermo Fisher Scientific, USA).

### Heat flow cell construction and setup

200 $\mu$m thick FEP film (Holscot, The Netherlands) was cut into the designed microfluidic shape with an industrial plotter (CE6000-40 Plus, Graphtec, Germany) and sandwiched between two sapphires (Kyburz, Switzerland) of thickness 500 $\mu$m (cooled sapphire, with four laser-cut holes of 1 mm diameter) and 2000 $\mu$m (heated sapphire, without holes). The sapphire-FEP-sapphire sandwich was then placed on an aluminum base, with an additional layer of heat-conductive graphite foil (EYGS091203DP, 25 $\mu$m, 1600 W mK$^{-1}$, Panasonic, Japan) between the aluminum base and the sandwich, and fixed there with a steel frame and torque-controlled screws for homogeneous force distribution. A second heat-conductive graphite foil (EYGS0811ZLGH, 200 $\mu$m, 400 W mK$^{-1}$, Panasonic, Japan) ensured the thermal connection between the heated sapphire and the electrical heating element,

which was again connected with torque-controlled steel screws. The thickness of the microfluidic chamber was then measured using a confocal micrometer (CL-3000 series with CL-P015, Keyence, Japan). Next, the chamber was pre-flushed with low-viscosity fluorinated oil (3 M™ Novec™ 7500 Engineered Fluid) to check for tightness and to drive out gas inclusions. The assembled chamber was mounted (with an intermediate layer of 200 μm heat conducting graphite foil, see above) onto an aluminum block which itself was connected to a cryostat (Grant R5 and TXF200, Grant Industries, UK) for cooling. The heaters were connected to a 400 W 24 V power supply and solid-state relays controlled by Arduino boards running the open-source Repetier firmware. For details on optimizing the individual elements and construction, see also refs. [35],[45].

### Heat flow cell experiments (general)

The inputs and outputs of the heat flow cell were connected via tubings to syringes placed on high-precision syringe pumps (neMESYS 290N low-pressure syringe pump with low-pressure quad syringe holder, Cetoni, Germany). The following microfluidic connections were used (all Techlab, Germany): Connectors (Connector inch, UP P-702-01), End Caps (Tefzel cap for 1/4-28 nut, UP P-755), Screws (Nut, Delrin, flangeless, VBM 100. 823-100.828), Ferrules (Ferrule VBM 100.632) and Tubing (Tubing Teflon (FEP), KAP 100.969). The chemically resistant syringes used were acquired from Göhler HPLC syringes, Germany: 2606714, 2606814, 2606914, 2606015, 2606035, 2606055, and 2606075 (ILS, Germany). The cryostat was set to −30 °C and the heating elements to 95 °C, resulting in temperatures of 20 °C and 50 °C on the cold and hot sides of the sapphire microfluidic chamber, respectively and translating to a temperature difference of 20 K between the inner surfaces of the sapphires[35]. Temperatures were measured on the respective outer surfaces of the sapphires using a thermal imaging camera (ShotPRO thermal imaging camera, Seek Thermal, USA). Before starting the experiments, all tubes and the thermal chamber were rinsed with fluorinated oil, and samples were loaded into the inlet tubes.

### Heat flow cell experiments (separation, Fig. 2)

For separation, an inflow of 15 nl s⁻¹ was applied, and the flow rates of the syringe pumps (controlled via neMESYS UserInterface, cetoni, Germany) were selected such that 5 % of the inflow was taken from the lower outlet and 95 % from the upper outlet. The experiments were run for 3 days. Stopping experiments was achieved by setting applied temperatures to room temperature. Samples were then removed from the tubings and collected for ion chromatographic measurement and further experiments.

### Heat flow cell experiments (up-concentration, Fig. 3)

In order to start experiments, the lower outlet was closed, and an inflow of 30 nl s⁻¹ was applied with an applied temperature gradient as described above. The experiments were run for 1 week. The experiments were stopped by setting the applied temperatures to room temperature and disconnecting the tubings. The chamber was then frozen at −80 °C for at least 1 h. By opening the frozen chamber and sequentially melting 25 % fractions from bottom to top (also see ref. [45]), local concentrations were measured by ion chromatography. The pore concentration (Fig. 2) is the average of all four fractions.

### Ion chromatography

Samples were injected using an autosampler (AS-DV, ThermoFisher Scientific, USA) and simultaneously measured in two ion chromatography systems.

Measurement of cations was done using an ion chromatography system (Dionex Aquion, ThermoFisher Scientific, USA) with an analytical column (Dionex IonPac CS12A), guard column (Dionex IonPac

CG18) and suppressor (Dionex CDRS 600). The chromatography method was set to provide 0.25 ml min⁻¹ flow using isocratic elution with 20 mM MSA, 15 mA suppression, a cell temperature of 40 °C, and a column temperature of 35 °C. Eluted ions were detected with a conductivity detector (DS6 Heated Conductivity Cell).

Measurement of anions was done using a separate ion chromatography system (Dionex Integrion, ThermoFisher Scientific, USA) with an analytical column (Dionex IonPac AS16 2 mm), guard column (Dionex IonPac AG16 2 mm), suppressor (Dionex ADRS 600 2 mm), eluent generator (EGC 500 KOH) and trap column (Dionex CR-ATC 600). Here, the method comprised a gradient elution starting with 57.5 mM KOH (for 10 min), a linear increase to 62.5 mM KOH over 2 min, isocratic elution with 62.5 mM KOH for 5 min, a direct step to 57.5 mM, and equilibration for 8 min. The flow was set to 0.25 ml min⁻¹, with the suppression current set to 47 mA. The cell temperature was set at 40 °C and the column temperature at 35 °C. Eluted anions were measured with a conductivity detector (DS6 Heated Conductivity Cell).

Data was analyzed using Chromeleon 7.2.10 (ThermoFisher Scientific, USA). Calibration was done using standard solutions. For exemplary chromatograms, see Supplementary Fig. 11.

### pH neutralization of solution

For each re-neutralization step towards a target $pH_{target}$, the pH of the solution $pH_{sol}$ was measured, and the following concentration of NaOH was added:

$$c_{NaOH} = c_{NaOH}^{pH} + c_{NaOH}^{pK_{a1}} + c_{NaOH}^{pK_{a2}} \qquad (1)$$

with

$$c_{NaOH}^{pH} = 10^{-pH_{sol}} + 10^{-(14-pH_{target})} \qquad (2)$$

as compensation of the pH difference and

$$c_{NaOH}^{pK_a} = c_{PO_4} * \left( \frac{10^{pH_{target}-pK_a}}{1+10^{pH_{target}-pK_a}} - \frac{10^{pH_{sol}-pK_a}}{1+10^{pH_{sol}-pK_a}} \right) \qquad (3)$$

to overcome the $pK_a$s of phosphoric acid at 2.2 ($pK_{a1}$) and 7.2 ($pK_{a2}$). Phosphate concentrations were measured using ion chromatography (see above).

We incubated all samples after pH adjustment for 2 weeks at 60 °C to reach the precipitation equilibrium. As this process leads to a shift in the pH, we repeated this procedure until no further precipitation occurred at a static pH. Three neutralization steps proved sufficient in the experiments, validated by modeling (Fig. 2d, Supplementary Fig. 4).

### Geochemical modeling

PHREEQC[52] was used for modeling, both in its standalone version (3.7.3) and in phreeqpython from Vitens[53]. The code is supplied in Supplementary Data 1.

### SEM-EDX analysis of precipitates and of geomaterials

Particles were carbon coated and measured with a Hitachi SU-5000 SEM (Japan). Samples were imaged in Back-Scattered Electron mode in a high vacuum and elements were measured by Energy Dispersive X-ray spectroscopy (EDX). Acceleration voltage was set to 15 and 20 kV, Working Distance (WD) -10 mm, at 240,000 nA emission current. Element mappings were performed over selected areas with additional point measurements to distinguish between individual crystals.

## Synthesis experiments of trimetaphosphate

Experiments were done in glass vials (17374073, ThermoFisher, USA) which were filled with 10 μl of sample obtained after pH neutralization (see above paragraph "pH neutralization of solution") of the inlet sample and the bottom/ΔT-labeled outlet sample of the heat flow cell experiments shown in Fig. 2 and heated to 180 °C in an oven (Memmert UNB 100 Oven, Germany). After 3 days, the vials were retrieved, cooled to room temperature, eluted in 500 μl ion chromatography water, and measured by ion chromatography. In the experiments shown in Supplementary Fig. 10, 30 mg of geomaterial was added to the vials before heating with an otherwise identical protocol.

## Other geomaterial samples

The geomaterial used in Fig. 3a, as well as Supplementary Fig. 10, is abbreviated as follows. BF2: Basalt F2 sample from Kilauea volcano in Hawaii. ILL: Illite. KAO: Kaolinite clay, purchased from Sigma Aldrich (USA). MON: Montmorillonite clay, purchased from Sigma-Aldrich (USA). ZEO: Zeolite, obtained from Zeocem (Slovakia). CAS: Carbonate sand, a mixture of carbonate ooids and other grains that were collected in the Bahamas, containing fragments of foraminifera and other carbonate shells. BSS: Basalt sand, crushed up, iron-rich basalt. SCS: Siliciclastic sand, quartz-rich beach sand containing iron minerals. VCG: Volcanic glass, crushed and powered obsidian. Compositions of the geomaterials were determined by SEM-EDX (see paragraph "SEM-EDX analysis of precipitates and of geomaterials") and are listed in Supplementary Table 2. The corresponding SEM images are shown in Supplementary Fig. 7-8.

## Conversion of trimetaphosphate (TMP) from initial acidic-dissolved phosphate

To obtain the conversion to TMP of the initially acidic-dissolved phosphate, we calculated:

$$\text{conversion} = 3 * c_{TMP} / c_{PO_4}^{pH_{acidic}} \tag{4}$$

(as each TMP contains three phosphates).

## Leaching experiments (Supplementary Figs. 1, 2 and Table 1)

Samples were weighed and mixed with 150 μl of ion chromatography water of the chosen pH. In contrast to the solution used in the heat flow cell experiments, the pH was not re-adjusted, exploring the amount of phosphate (and other salts) leached upon incubation at a given initial pH. After initial vortexing, leaching took place at a controlled temperature (T100 Thermal Cycler, Bio-Rad Laboratories). After the experimental incubation time, the samples were cooled to room temperature, vortexed, and centrifuged. The particle-free supernatant was then diluted with IC grade water for IC measurement.

## Data availability

All data are available in the main text or the supplementary materials. The data generated in this study are provided in the Source Data. Source data are provided with this paper.

## Code availability

The code used to simulate precipitation dynamics shown in Fig. 2 and Supplementary Fig. 4 is supplied in Supplementary Data 1.

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

## Acknowledgements

The authors thank P. Aikkila for experimental support and P. Aikkila, I. Smokers, A. Dass and J. Langlais for fruitful discussions. This work was funded by the Deutsche Forschungsgemeinschaft (DFG, German Research Foundation) under Project-ID 364653263—CRC 235 (T.M., A.S., D.W., D.B.D., B.S., D.B., C.B.M.), under Project-ID 521256690—CRC 392 (T.M., D.W., B.S., D.B., C.B.M.), under Project-ID 201269156—SFB 1032 (M.R., D.B.) and under Germany's Excellence Strategy EXC-2094-390783311 (T.M., A.S., D.B., C.B.M.). Funding from the Volkswagen Initiative 'Life?—A Fresh Scientific Approach to the Basic Principles of Life' (T.M., A.Z.C., D.B.D., D.B., C.B.M.) is gratefully acknowledged. The work is supported by the Center for Nanoscience Munich (CeNS).

## Author contributions

Conceptualization and Design: T.M., A.S., C.B.M.; Methodology: T.M., A.S., D.B., C.B.M.; Experiments: T.M., A.S., M.R., D.W.; Data analysis: T.M., A.S., M.R., D.W., K.K.; Modeling: T.M., M.R., C.B.M.; Provision of geomaterial: A.Z.C., K.R.M.; Visualization: T.M., C.B.M.; Funding acquisition: D.B.D., B.S., D.B., C.B.M.; Supervision: CBM; Writing—original draft: T.M., C.B.M.; Writing—review & editing: T.M., A.S., M.R., D.W., A.Z.C., K.R.M., T.B., D.B.D., K.K., F.G., B.S., D.B., C.B.M.

## Funding

## Competing interests

The authors declare no competing interests.
