## [Transparent Peer Review file · Nature Communications]

Heat flows solubilize apatite to boost phosphate availability for prebiotic chemistry

Corresponding Author: Dr Christof Mast

Version 0:

Reviewer comments:

Reviewer #1

(Remarks to the Author)

Non-equilibrium conditions such as chemical and thermal gradients could have contributed to the emergence of life. Previously studies have focused on the catalytic properties of minerals for the synthesis of prebiotically relevant molecules. In this paper, the authors raised an interesting question on the source of phosphate given its low concentration in the earth's crust and typically forming solid precipitates that hinders its availability to produce life's building block molecules such as nucleobase and trimetaphosphate (TMP). Following their previous studies on selective magnesium up-concentration and enrichment of certain prebiotic molecules, these authors demonstrate that heat flows offer additional benefits over the bulk condition including localized concentration of phosphate that could be readily converted to TMP under high temperatures. The experiments were well-designed, and the manuscript is commendably structured, supported by robust data, and articulately presented. I recommend its publication in Nature Communications suggesting minor revisions to further refine its content and improve clarity.

1. The first two paragraphs of the Results section was not assigned a subtitle like the others. Figs. 2B-D were mentioned before Fig. 2A was introduced.
2. The experiments in Fig. 2 were at pH 1.6-4.0. Why did the author select this particular range and the individual pH values?
3. The authors observed over 100-fold up-concentration under heat-flow conditions. Are these results time-dependent? Is there a time threshold for such effect to be observed? Will this concentration effect become stronger if the experiments were left even longer?
4. A flow rate of 30 nl/s was used for the accumulation experiments. Including schematics of the microfluidic channel with dimension details would help the reader to understand how fast the flow is. It would also help clarify how spatially distant the so-called "niche", "pore", and "bulk" are.
5. The heat-flow experiments were performed with acidic-dissolved apatite solution. In realistic settings, there could be apatite (or other mineral) particles flowing around the pores. Perhaps, the pores themselves consist of apatite. Could the authors discuss the possible effects of the presence of these minerals on the effectiveness of selective phosphate concentration and accumulation?

Reviewer #2

(Remarks to the Author)

Manuscript entitling, "Heat flows solubilize apatite to boost phosphate availability for prebiotic chemistry" by Matreux et al. was an interesting paper based on the 'Phosphate Problem' in the realm of prebiotic phosphorus chemistry. Apatite being the major carrier of phosphate on the early Earth and being practically insoluble insert portrayed a challenging issue to overcome. To answer this issue, the author presented the work given in the present study. While, I was very interested in studying the research work, I was somewhat disappointed to find out that the current submitted manuscript seemed to lack important details/questions and clarity at times. I am going to point out only some of the major issues:

1. Line 101, I am not sure why trimetaphosphate (TMP) has been referred as 'reactive' throughout the manuscript. Such

terms should be avoided without implying properly.

2. Figure 1 seemed to be misleading as it also pointed out towards the phosphorylation process/nucleotide formation. Since no reaction studied between the organic-raw solutions mixtures containing dissolved/leached phosphate were carried out, such illustrations could be misleading.

3. Line 123, it was not clear as to why the three locations mentioned e.g., Ipirá complex, Brazil, Durango, Mexico and Ontario, Canada were chosen? Was there any significance, something pertaining to pH, temperature or any other conditions?

4. Line 127, it would be significant to give the actual value of the orthophosphate concentration rather than mentioning previous results here. In fact this manuscript suffered from the lack of any table provided showing the experimental details/conditions, and the amount of phosphate leached/released in each experiment for a fair comparison. Furthermore Line 141, heating at 180 °C cannot be considered as 'mild'.

5. While it was highly relevant and significant to the prebiotic community to present the result of the release of orthophosphate minerals at neutral pH which is highly significant, however, many experiments did show the better release of orthophosphate under lower pH values. It should be noted that apatites become soluble at lower pHs is a well-known phenomenon (also see Adcock, C., Hausrath, E. & Forster, P. Readily available phosphate from minerals in early aqueous environments on Mars. *Nature Geosci* 6, 824–827 (2013). <https://doi.org/10.1038/ngeo1923>).

6. No diagrammatic illustration of the heat-flow equipment was provided.

7. Results section, once again as mentioned above, there should have been a table providing the concentrations/ or yields (%) (ideally both) for each one of the experiments and whether TMP was obtained or not.

8. It was not explored as to why TMP was the only potential polymerization product of the heating reaction was more simpler or more complex P-species were also observed such as pyrophosphate (simpler), and tetra-phosphate (more complex). Here, the manuscript seemed to be greatly suffered from a qualitative ³¹P-NMR that would have shown potentially all P species without the need of any standard materials available. Not only, it would be easy to show various P species, ideally the TMP (product) could also be spiked by the standard compound to exactly match and confirm. The manuscript was further adversely impacted by the fact that there was no chromatograms of any sort were provided to show the P contents in each sample.

9. Experimental section, 'synthesis of TMP' it was mentioned as 'sample was heated' what sample?

10. Leaching experiments study, SI (Figure 1) caption stated "Washing of grains with H₂O₂ does not change the composition and concentration of leachates either. It was not clear why H₂O₂ would be necessary for this purpose, as H₂O₂ can quickly alter the pH and can cause oxidation reaction onto the surfaces how can it be claimed without an evidence that the above mentioned statement stating that the washing of sample grains with H₂O₂ does not impact..... Also what was the concentration of the peroxide used?

11. In case of clay/minerals used as geo-materials, no attempts/experiments were suggested to show if there was any interactions between the orthophosphate and cations/anions found in the compositional units of these geo-materials. Also the supplementary section Figures 5 and 6 showing the SEM images of the starting materials were found to be of least significant as such data are already accessible. These SEM figures could be more significant if 'before' and 'after' reactions the changes were observed.

12. Polymerization set up in the Figure 7 failed to set a strong case as it employed NaH₂PO₄. It was not clear why would sodium phosphate be used here? Why not apatite/ leached solution? Also similar reactions have already been reported (see: Keefe AD, Miller SL. Are polyphosphates or phosphate esters prebiotic reagents? *J Mol Evol*. 1995 Dec;41(6):693-702. doi: 10.1007/BF00173147).

I strongly suggest to authors to completely rewrite the manuscript and add all the significant information that this current manuscript lacks including the chromatograms, proof of TMP in solutions, comprehensive tables providing all the details of experiments as well as yields (or concentrations) of phosphate, side by side to highlight the significance of this manuscript.

Version 1:

Reviewer comments:

Reviewer #1

(Remarks to the Author)

The authors have addressed all reviewers' comments properly as well as added significant details and corrections to the manuscript. Therefore, I recommend this manuscript for publication in *Nature Communications*.

Reviewer #2

(Remarks to the Author)

I thank the authors for taking time to address all the comments and concerns raised. Although, the authors have satisfactorily revised the manuscript and most of the raised concerns, I would still like to comment that the point No 5, raised in my previous round of revisions is not satisfactorily addressed. My concern is/was, " While it was highly relevant and significant to the prebiotic community to present the result of the release of orthophosphate minerals at neutral pH which is highly significant, however, many experiments did show the better release of orthophosphate under lower pH values. It should be noted that apatites become soluble at lower pHs is a well-known phenomenon (also see Adcock, C., Hausrath, E. & Forster, P. Readily available phosphate from minerals in early aqueous environments on Mars. *Nature Geosci* 6, 824–827 (2013). <https://doi.org/10.1038/ngeo1923>)".

I still think that as such the manuscript lacks novelty and its results are very much as expected for apatite to release dissolve phosphate under acidic conditions as also supplemented by the previous studies by Adcock et al. 2013. It came to me as no surprise or of novelty that the experimental pH range was 1.6-4.

The authors also mention in their referees comments No 2 " Under more alkaline conditions $\text{pH} > 4$, the release of phosphate is more than 3 orders of magnitude lower compared to acidic conditions, so that the concentration achievable by the heat-flow-induced separation of calcium and phosphate decreases equally. For the thermal disequilibrium to enable sufficiently high phosphate concentrations for prebiotic chemistry (10–100 mM), acidic conditions must have existed beforehand for apatite dissolution....."

All in all, in my opinion, most of the concerns have been successfully addressed by the authors, although I still question the novelty of the work due to above mentioned reason and do not consider this as such a novel idea.

Answer to referee comments are **green**. Changes in the manuscript text are **blue**.

REVIEWER COMMENTS

Reviewer #1 (Remarks to the Author):

Non-equilibrium conditions such as chemical and thermal gradients could have contributed to the emergence of life. Previously studies have focused on the catalytic properties of minerals for the synthesis of prebiotically relevant molecules. In this paper, the authors raised an interesting question on the source of phosphate given its low concentration in the earth's crust and typically forming solid precipitates that hinders its availability to produce life's building block molecules such as nucleobase and trimetaphosphate (TMP). Following their previous studies on selective magnesium up-concentration and enrichment of certain prebiotic molecules, these authors demonstrate that heat flows offer additional benefits over the bulk condition including localized concentration of phosphate that could be readily converted to TMP under high temperatures. The experiments were well-designed, and the manuscript is commendably structured, supported by robust data, and articulately presented. I recommend its publication in Nature Communications suggesting minor revisions to further refine its content and improve clarity.

We thank the referee for this positive assessment of the work and for emphasizing the importance of the question for research into the emergence of life.

1. The first two paragraphs of the Results section was not assigned a subtitle like the others. Figs. 2B-D were mentioned before Fig. 2A was introduced.

We would like to thank the referee for these hints. We now refer to Figure 2a in the correct order and write in the main text (lines 170-172):

“To study this scenario experimentally (Fig. 2a), we prepared 5 samples in which ~3 g of apatite (Brazil apatite, see Table 1) was dissolved by repeatedly supplying HCl to reach the respective predefined initial pH values (shown in Fig. 2b-e).”

We further added a subtitle to the first two paragraphs of the Results section (line 120):

“**Apatite solubilization under equilibrium conditions**”

2. The experiments in Fig. 2 were at pH 1.6-4.0. Why did the author select this particular range and the individual pH values?

As the referee correctly points out, the reason for the choice of pH range is indeed insufficiently explained. We chose this pH range as in separate leaching experiments on various apatites using solutions with different initial pH values (pH 1-12), we detected usable concentrations of phosphate and calcium only in the range of pH~1-4, which is in agreement with the literature (main text reference 17, Table 1, Supplementary Figure 2a-c).

Under more alkaline conditions $\text{pH} > 4$, the release of phosphate is more than 3 orders of magnitude lower compared to acidic conditions, so that the concentration achievable by the heat-flow-induced separation of calcium and phosphate decreases equally. For the thermal disequilibrium to enable sufficiently high phosphate concentrations for prebiotic chemistry (10–100 mM), acidic conditions must have existed beforehand for apatite dissolution - a valid assumption, as described in references 18-24.

To make this clearer in the manuscript, we now write in lines 128-138:

“We exposed them to a wide range of pH conditions between pH 1 and pH 12. After reaching an equilibrium in pH and solute concentrations (Supplementary Fig. 2), we found that the amount of leached phosphate was largely consistent across the different samples and in agreement with previous literature studies¹⁷. The results confirmed the characteristic strong dissolution at acidic pH (see Table 1) with up to 5 mM leached phosphate concentration at pH 1, but low phosphate concentrations of approximately 10 μM at more neutral pH values compatible with prebiotic chemistry. In the following experiments, we focused on the pH range from 1.6 to 4, as found in acidic geothermal flows¹⁸⁻²⁴. More alkaline conditions would only trigger the marginal release of phosphate from apatite as shown in Supplementary Fig. 2a-c and visible in the drop of phosphate dissolution at pH 4 in Fig. 2b. “

Without this heat-flow-driven phosphate enrichment, the phosphate in solution would bind with calcium again under more neutral conditions, as would be required for prebiotic chemistry, and thus become unavailable.

To highlight this more clearly in the manuscript, we have added lines 203-208:

“

We explored the same pH conditions between pH 1.6 and pH 4 for apatite dissolution, as in the equilibrium case above, to study the respective characteristics of phosphate enrichment. While without thermal gradients the near-surface neutralization of the dissolved apatite samples leads to a re-precipitation of phosphate with calcium, we hypothesized that the heat-flow-driven fractionation of the two ions may result in high phosphate concentrations remaining in solution even after pH neutralization.

“

3. The authors observed over 100-fold up-concentration under heat-flow conditions. Are these results time-dependent? Is there a time threshold for such effect to be observed? Will this concentration effect become stronger if the experiments were left even longer?

The referee is right that the accumulation process is indeed time dependent. To illustrate the behavior of phosphate accumulation over longer periods, we calculated its time evolution in a numerical simulation in the newly added Supplementary Figure 9 over 100 days. As can be seen there, the phosphate concentration even increases up to 1000 times the initial concentration and would continue to rise according to (Mast et al, 2013, see below) below. However, since the strength of thermophoresis, or the Soret coefficient, depends on the ionic strength of the surrounding medium (Reichl et al, 2014, see below), this heat-flow-driven accumulation would regulate itself from a certain level. For magnesium ions with Soret coefficients in the same range, no possibly detrimental concentration dependence of the accumulation process has been found even at $[Mg^{2+}] = 40$ mM, so that the same can be approximately assumed for the 65 mM phosphate accumulated here after one week.

To elaborate on this, we now write in the discussion in lines 307 to 312:

“As illustrated in Supplementary Fig. 9, these high phosphate concentrations reached after one week do not yet reflect the final steady state of the system, but will increase to 1000-fold after 100 days. Due to the robustness of such systems to geometrical⁵⁸ or temporal⁴⁴ perturbations, significantly higher concentrations could be achieved^{59,60}, limited only by the duration of the accumulation process and the ionic strength of the solvent⁶¹. “

References:

Christof B. Mast, Severin Schink, Ulrich Gerland and Dieter Braun

“Escalation of Polymerization in a Thermal Gradient”

PNAS 110, 8030-8035 (2013)

Maren Reichl, Mario Herzog, Alexandra Götz and Dieter Braun

“Why charged molecules move across a temperature gradient: the role of electric fields”

Physical Review Letters 112, 198101 (2014)

4. A flow rate of 30 nl/s was used for the accumulation experiments. Including schematics of the microfluidic channel with dimension details would help the reader to understand how fast the flow is. It would also help clarify how spatially distant the so-called “niche”, “pore”, and “bulk” are.

Following the referee's comment, we now provide a sketch of the experimental setup in Supplementary Figure 6 with the corresponding subdivision into “niche”, the lowest quarter = 12.5mm of the chamber, as well as “pore”, i.e. the entire chamber over 50mm in height. “bulk” corresponds to the reservoir, the diluted phosphate solution, which flows into the chamber at 30nl/s. To be able to better estimate the resulting flow velocities, we have shown different cross-sections in the chamber entrance and chamber center in the new Supplementary Figure 9b). As shown there, the flow velocities generated by the flow disturb the thermal convection, and thus the heat-flow-driven accumulation only in the uppermost 4 mm of the chamber, so that efficient phosphate concentration is readily possible.

In the text, we refer to the new supplementary figures and describe the different sections of the heat-flow chamber in more detail in lines 272 to 280:

“We speculated that the closed pore, which lacks a lower material outflow, would maximize the heat-flow-driven phosphate accumulation. For each experiment at a defined pH value, we applied a volume flow rate of 30 nl/s, fed by reservoirs of this pH, each with 500 μ M dilute phosphate over the course of one week (“bulk”, Fig. 3b). While these flow rates generated flow velocities of about 160 μ m/s in the inlet channels, they only influenced the convection currents in the uppermost 4 mm of the heat-flow chamber, so that an efficient accumulation of the phosphate could take place (Supplementary Fig. 9b). Due to the high volumes required for these experiments, we used freshly dissolved NaH_2PO_4 as a representative soluble phosphate.

After the experiment, we froze the entire chamber and segmented the frozen content along the height of the chamber into four fractions of equal volume. IC measurements of all fractions revealed an up to 130-fold concentration increase of orthophosphate in the lowest quarter fraction with a height of 12.5 mm (Fig. 3b “niche”, Supplementary Fig. 9), corresponding to an absolute phosphate concentration of 65 mM, with an up to 40-fold boost of phosphate averaged over the entire pore (Fig. 3b “pore”). “

5. The heat-flow experiments were performed with acidic-dissolved apatite solution. In realistic settings, there could be apatite (or other mineral) particles flowing around the pores. Perhaps, the pores themselves consist of apatite. Could the authors discuss the possible effects of the presence of these minerals on the effectiveness of selective phosphate concentration and accumulation?

The referee correctly points out that in a realistic geological system, the processes of phosphate leaching and heat-flow-driven fractionation could not occur separately but simultaneously due to the geomaterial being the natural boundaries of the heat-flow chamber. In our experiments, both processes were carried out sequentially for better controllability, however, this approach does also represent a realistic approximation of natural systems: Since the leaching processes occur relatively quickly and the heat-flow chambers do not occur as single entities, but are likely to be found in nature as larger systems of interconnected rock cracks, the dissolution equilibrium in such systems would have been established preceding the situation explored here.

We now discuss this in more detail in lines 315-323:

“In this work, the phosphate leaching from geomaterial and the phosphate enrichment driven by the heat flow were carried out sequentially to maximize reproducibility and to clarify the key mechanisms. In a natural system, the walls of heat-penetrated rock cracks or sedimented grains themselves could contain phosphate-bearing material, so that both processes could, in principle, take place simultaneously. The resulting more complex crack shapes do not hinder accumulation⁵⁸ and the comparatively fast leaching process of apatite at the low pH values considered here at rates¹⁷ of $10^{-8} \text{ mol/m}^2\text{s}$ would provide continuous feeding and drive further accumulation in downstream chambers on which we focused in our experimental model.“

Reviewer #2 (Remarks to the Author):

Manuscript entitling, “Heat flows solubilize apatite to boost phosphate availability for prebiotic chemistry” by Matreux et al. was an interesting paper based on the ‘Phosphate Problem’ in the realm of prebiotic phosphorus chemistry. Apatite being the major carrier of phosphate on the early Earth and being practically insoluble insert portrayed a challenging issue to overcome. To answer this issue, the author presented the work given in the present study. While, I was very interested in studying the research work, I was somewhat disappointed to find out that the current submitted manuscript seemed to lack important details/questions and clarity at times. I am going to point out only some of the major issues:

1. Line 101, I am not sure why trimetaphosphate (TMP) has been referred as ‘reactive’ throughout the manuscript. Such terms should be avoided without implying properly.

We thank the referee for this comment and agree that the term “reactive” could be misleading here. In the updated manuscript, we have now corrected this and describe TMP as an important phosphorylating and activating agent for prebiotic chemistry. For example, we now write in lines 303-305:

“We demonstrated that the achieved phosphate concentrations of up to 15 mM strongly enhanced the formation of TMP, an important activating and phosphorylation agent for driving various prebiotic reactions.”

2. Figure 1 seemed to be misleading as it also pointed out towards the phosphorylation process/nucleotide formation. Since no reaction studied between the organic-raw solutions mixtures containing dissolved/leached phosphate were carried out, such illustrations could be misleading.

In fact, the representation of RNA structures in Figure 1a may be seen as somewhat misleading – the intention here was to highlight the prominent role of phosphate in important biopolymers. We have improved this in an updated figure and now only indicate the structure of TMP, which is synthesized from the phosphate solubilized from apatite, as shown in Figure 2, and which serves as an important activator for a variety of reactions considered prebiotically relevant. We added the figure titles to Fig. 1a “Phosphate problem” and to Fig. 1b “Heat-flow-driven P-solubilization” to clarify the distinction between the problem and the motivation of the work (a) and the proposed non-equilibrium approach as its solution (b).

3. Line 123, it was not clear as to why the three locations mentioned e.g., Ipirá complex, Brazil, Durango, Mexico and Ontario, Canada were chosen? Was there any significance, something pertaining to pH, temperature or any other conditions?

We thank the referee for pointing this out – in fact, the reason for using apatites from three different origins was that the apatites used are representative samples, but so far this has not been made entirely clear. We wanted to test more than one apatite mineral as natural minerals show natural heterogeneities in their composition.

We chose to work with Durango Apatite (Mexico) as in geosciences this Apatite serves as standard for microprobe analysis and is as well introduced as standard for LA-ICPMS analysis (e.g. Chew et al. 2016, see below). As it is literally impossible to receive larger quantities of this apatite, we chose the apatite from the Ipirá complex (Brazil) as here homogeneous samples in the order of several 100g are available (not as idiomorph crystals but as massive blocks). The apatite from Ontario (Canada) was added to the analysis as it has a subtly different composition (Cl content is below detection limit of the XRF analysis and its F content is the lowest and its CaO content the highest in all 3 fluorapatites). To better illustrate this, we have created a new Supplementary Figure 1, which summarizes the apatite compositions and phosphate concentrations from the corresponding leachates and shows that the different samples show similar leachate compositions despite their slightly different oxide compositions and different collection sites.

We have added Table 1 in the main text to emphasize this and have now explained this in more detail in the main text on lines 124-131:

“To obtain an experimental baseline without the effect of heat-flow-driven enrichment, we first characterized the amount of phosphate leached from three natural apatite samples under bulk conditions^{17,35} (Ipirá Complex, Brazil, Durango, Mexico and Ontario, Canada, see Table 1 and Supplementary Fig. 1). We exposed them to a wide range of pH conditions between pH 1 and pH 12. After reaching an equilibrium in pH and solute concentrations (Supplementary Fig. 2), we found that the amount of leached phosphate was largely consistent across the different samples and in agreement with previous literature studies¹⁷.“

Reference:

Chew, David M., Michael G. Babechuk, Nathan Cogne, Chris Mark, Gary J.

O'Sullivan, Isadora A. Henrichs, Daniel Doepke, and Cora A. McKenna.

"(LA, Q)-ICPMS trace-element analyses of Durango and McClure Mountain apatite and implications for making natural LA-ICPMS mineral standards."

Chemical Geology 435: 35-48. (2016)

4. Line 127, it would be significant to give the actual value of the orthophosphate concentration rather than mentioning previous results here. In fact this manuscript suffered from the lack of any table provided showing the experimental details/conditions, and the amount of phosphate leached/released in each experiment for a fair comparison.

Following the referee's suggestion, we have added more excerpts of absolute concentration values from the figures 2-3 and the Supplementary Figures to the text. For example, in lines 132-134:

“The results confirmed the characteristic strong dissolution at acidic pH (see Table 1) with up to 5 mM leached phosphate concentration at pH 1, but low phosphate concentrations of approximately 10 μ M at more neutral pH values compatible with prebiotic chemistry.”

In lines 210-212, we added:

“As depicted in Supplementary Fig. 3, in the bottom outflow, phosphate was accumulated up to 70 % stronger than calcium, reaching absolute concentrations of 50 mM, compared to 53 mM of calcium (see Table 2).”

The values used for the figures in the main text are now also attached as xls file in the Supplementary Data with all experimental details, which also contain the integral values from the ion chromatography measurements for the samples. We also include the reaction yield for the synthesis of polyphosphates for the experiments shown in Figure 2e in these tables. As an example, we have also added some representative ion chromatograms in the Supplementary Figure 11, from which this data was obtained.

Furthermore Line 141, heating at 180 °C cannot be considered as ‘mild’.

We thank the referee for this comment. Mild was somewhat misleading in this context - it referred to the comparison with the TMP synthesis, e.g. under volcanic conditions and at much higher temperatures (10.1007/bf01804669). Following this, we have replaced the three occurrences of “mild” with the more appropriate term “moderate”, as is already used in the literature for this temperature range (e.g. 10.1038/s43247-024-01657-4), and have now added the corresponding references.

5. While it was highly relevant and significant to the prebiotic community to present the result of the release of orthophosphate minerals at neutral pH which is highly significant, however, many experiments did show the better release of orthophosphate under lower pH values. It should be noted that apatites become soluble at lower pHs is a well-known phenomenon (also see Adcock, C., Hausrath, E. & Forster, P. Readily available phosphate from minerals in early aqueous environments on Mars. *Nature Geosci* 6, 824–827 (2013). <https://doi.org/10.1038/ngeo1923>).

The referee is right that apatite is easily soluble under acidic conditions. To emphasize this, we write in lines 58-61:

“While apatite easily dissolves under low pH conditions (1.5 – 2.5)^{13,17}, as found in lacustrine environments, acidic lakes^{18,19}, hot springs²⁰, and hydrothermal systems^{21–24} with steep pH gradients^{25–28}, such environments are often incompatible with prebiotic chemistry, which commonly requires neutral to alkaline conditions^{18,24,28}. “

in which we also reference the work mentioned by the referee. As stated there and in the following lines, the hunger of prebiotic chemistry for phosphate at neutral pH however poses a major problem.

“Here, the dissolved phosphate quickly precipitates with presumably abundant calcium, so that despite the broad range of pH conditions available in geological systems, one is back at square one of the phosphate problem.”

6. No diagrammatic illustration of the heat-flow equipment was provided.

We thank the referee for this suggestion and show the structure of the heat-flow chamber in the newly created Supplementary Figure 6, as well as the microfluidic partitioning, as used in Figure 3. This new figure is now referenced in the first description of the heat flow chamber in the main text on line 194.

7. Results section, once again as mentioned above, there should have been a table providing the concentrations/ or yields (%) (ideally both) for each one of the experiments and whether TMP was obtained or not.

Following the referee's comments, we have now created detailed data tables with all integration and concentration values for the various phosphate species as a Supplementary Data file or, in excerpts, as a Table 2 in the main text.

8. It was not explored as to why TMP was the only potential polymerization product of the heating reaction was more simpler or more complex P-species were also observed such as pyrophosphate (simpler), and tetra-phosphate (more complex). Here, the manuscript seemed to be greatly suffered from a qualitative ^{31}P -NMR that would have shown potentially all P species without the need of any standard materials available. Not only, it would be easy to show various P species, ideally the TMP (product) could also be spiked by the standard compound to exactly match and confirm. The manuscript was further adversely impacted by the fact that there was no chromatograms of any sort were provided to show the P contents in each sample.

In Supplementary Figure 3e-g and as part of the data tables, we now also show the concentration values for di- and triphosphates for the experiments described in Figure 2e. In addition, in Supplementary Figure 11 we now show examples of the ion chromatograms, which show the existence of the polyphosphates from the experiment in comparison to standards. Figure 2e focuses mainly on trimetaphosphate due to its importance as a phosphorylation agent for prebiotic chemistry. Together with the data now added for the polyphosphates, a more complete picture should emerge.

The referee rightly points out that ^{31}P -NMR is a valid method for the detection of polyphosphates. Since the thermal non-equilibrium microfluidics allows a volume flow of only up to 0.75nl/s at the lower chamber outlet that yields the phosphate-enriched solution (see Figure 2a, labeled ΔT and main text lines 192-200), only a total maximal volume of 190 μl of up to 50 mM can be collected even after three days of experimentation which is split among all subsequent experimental steps, e.g. intermediate pH and IC-measurement as well as the phosphate-polymerization. A longer runtime would reduce flow-precision due to repetitive refilling of the syringe pumps or using large syringes.

With the analytical workflow and observed reaction yields, this corresponds to final sub-millimolar concentration in 10 μl samples.

While this is not a drawback for natural systems due to their size and parallelizability, this sample amount is not sufficient for standard ^{31}P -NMR, as also shown in Supplementary Figure 11. The ion chromatography used here represents a suitable alternative for smaller sample amounts, the results of which we are now demonstrated with example chromatograms (Supplementary Figure 11). To emphasize this, we now write in lines 195-202:

“In this flow-through system, we set the inflow volume rate⁵³ of acidic-dissolved apatite solution to 15 nl/s. The outflow volume rates were set to 0.75 nl/s (bottom) and 14.25 nl/s (top). The experiment was run for a period of three days until sufficient sample volume was obtained for analysis using ion chromatography and for subsequent phosphate polymerization (see Fig. 2e and Supplementary Fig. 3e-g). Such moderate flow rates were shown to not disturb the selective accumulation of ions through concurrent solvent convection (black arrows) and solute thermophoresis (white arrows)^{27,35,44}. “

9. Experimental section, ‘synthesis of TMP’ it was mentioned as ‘sample was heated’ what sample?

We thank the referee for pointing out this omission. We now specify this in more detail and write in the Methods in lines 481-485:

“Experiments were done in glass vials (17374073, ThermoFisher, USA) which were filled with 10 μ l of sample obtained after pH neutralization (see above paragraph “pH neutralization of solution”) of the inlet sample and the bottom/ ΔT -labeled outlet sample of the heat flow cell experiments shown in Fig. 2 and heated to 180 °C in an oven (Memmert UNB 100 Oven, Germany).”

10. Leaching experiments study, SI (Figure 1) caption stated “Washing of grains with H₂O₂ does not change the composition and concentration of leachates either. It was not clear why H₂O₂ would be necessary for this purpose, as H₂O₂ can quickly alter the pH and can cause oxidation reaction onto the surfaces how can it be claimed without an evidence that the above mentioned statement stating that the washing of sample grains with H₂O₂ does not impact..... Also what was the concentration of the peroxide used?”

We thank the referee for pointing to this detail in our preparation and thus giving us the chance to clarify the sample preparation.

As the samples we used in our experiments were not contaminated with organics, we did not use H₂O₂-washing for the leaching and dissolution experiments.

However, as washing with H₂O₂ is widely used in geoscience to remove biological contaminations, we added a comparison experiment in order to quantify the effect of H₂O₂ on subsequently leached concentrations.

To follow the referee’s suggestion, we now also added the peroxide concentration to the caption to Supplementary Figure 2d. The statement that washing does not affect the composition of the leachate refers to the mostly equal concentrations of fluorine, calcium and phosphate ions in the leachate at pH 2 and pH 6, as shown in Supplementary Figure 2d. To describe this more clearly, we now say in the corresponding figure caption:

“(d) Effect of H₂O₂-washing (10% vol.) on leached concentrations compared to unwashed grains yields comparable concentrations of the fluorapatite-forming ions fluorine, calcium and phosphate.”

To emphasize this, we now write in lines 350-354:

“The H₂O₂-washing of samples for comparison with unwashed samples (see Supplementary Fig. 2d) was done by incubating samples in H₂O₂ (10 vol%) for seven days to remove biological contaminations, rinsing them with purified water, and drying for 24 h at 200 °C in a furnace (similar to³⁵).”

11. In case of clay/minerals used as geo-materials, no attempts/experiments were suggested to show if there was any interactions between the orthophosphate and cations/anions found in the compositional units of these geo-materials. Also the supplementary section Figures 5 and 6 showing the SEM images of the starting materials were found to be of least significant as such data are already accessible. These SEM figures could be more significant if ‘before’ and ‘after’ reactions the changes were observed.

We thank the referee for this comment. In fact, the SEM images in Supplementary Figures 7 and 8 were only added for an overview of surfaces and grain sizes, as they were obtained as part of the EDX measurement of the geomaterial compositions for Supplementary Table 3, to get an understanding of the phosphate leaching efficiency processes that lead to the results shown in Figure 3a. To make this clearer and to distinguish this data from the polymerization reactions shown in Supplementary Figure 10, we write in lines 247-248:

“SEM images of geomaterials before leaching are shown in Supplementary Fig. 7-8 alongside the initial compositions in Supplementary Table 3 obtained by corresponding EDX analysis.”

In this context, we now motivate the actual polyphosphate synthesis separately and more clearly on these geomaterials in lines 286–288:

“The geomaterial that served as phosphate source in Fig. 3a could also support the formation of polyphosphates (see Supplementary Fig. 10) and enhance the formation of TMP up to 5.2-fold on kaolinite.”

12. Polymerization set up in the Figure 7 failed to set a strong case as it employed NaH_2PO_4 . It was not clear why would sodium phosphate be used here? Why not apatite/ leached solution? Also similar reactions have already been reported (see: Keefe AD, Miller SL. Are polyphosphates or phosphate esters prebiotic reagents? *J Mol Evol.* 1995 Dec;41(6):693-702. doi: 10.1007/BF00173147).

Together with the last note and the text changes in lines 289–291, it is now more clearly stated that the phosphate polymerization with solubilized phosphate under neutral conditions, as shown in Supplementary Figure 10, takes place as part of the scenario illustrated in Figure 1b and in the context of Figure 3. The use of NaH_2PO_4 is meant to be representative of a water-soluble phosphate obtained, e.g., by the heat-flow-driven Calcium-removing processes described in this work after neutralization with NaOH or the leaching of a variety of geomaterials. However, since both experiments would need long timescales (>months) to produce the volumes of liberated phosphate required for the heat-flow-driven enrichment (Fig. 3b) and polymerization on geomaterials (Supplementary Fig. 10), equivalent amounts of freshly dissolved NaH_2PO_4 were used. To clarify this, we now write on lines 278-280:

“Due to the high volumes required for these experiments, we used freshly dissolved NaH_2PO_4 as a representative soluble phosphate. “

I strongly suggest to authors to completely rewrite the manuscript and add all the significant information that this current manuscript lacks including the chromatograms, proof of TMP in solutions, comprehensive tables providing all the details of experiments as well as yields (or concentrations) of phosphate, side by side to highlight the significance of this manuscript.

We thank the referee for the numerous and helpful comments, which led us to elaborate large parts of the manuscript in more detail and to add main text Table 2 and the ion chromatography measurement values as data tables together with Supplementary Figure 11 for example measurements. We believe that the comprehensibility of the manuscript has now been significantly improved and that the utility of heat-flow driven phosphate solubilization from apatite for prebiotic chemistry is more clearly presented.

Reviewer #1 (Remarks to the Author):

The authors have addressed all reviewers' comments properly as well as added significant details and corrections to the manuscript. Therefore, I recommend this manuscript for publication in Nature Communications.

We thank the referee for this positive assessment and the recommendation for publication.

Reviewer #2 (Remarks to the Author):

I thank the authors for taking time to address all the comments and concerns raised. Although, the authors have satisfactorily revised the manuscript and most of the raised concerns, I would still like to comment that the point No 5, raised in my previous round of revisions is not satisfactorily addressed. My concern is/was, " While it was highly relevant and significant to the prebiotic community to present the result of the release of orthophosphate minerals at neutral pH which is highly significant, however, many experiments did show the better release of orthophosphate under lower pH values. It should be noted that apatites become soluble at lower pHs is a well-known phenomenon (also see Adcock, C., Hausrath, E. & Forster, P. Readily available phosphate from minerals in early aqueous environments on Mars. *Nature Geosci* 6, 824–827 (2013). <https://doi.org/10.1038/ngeo1923>)".

I still think that as such the manuscript lacks novelty and its results are very much as expected for apatite to release dissolve phosphate under acidic conditions as also supplemented by the previous studies by Adcock et al. 2013. It came to me as no surprise or of novelty that the experimental pH range was 1.6-4.

The authors also mention in their referees comments No 2 " Under more alkaline conditions $\text{pH} > 4$, the release of phosphate is more than 3 orders of magnitude lower compared to acidic conditions, so that the concentration achievable by the heat-flow-induced separation of calcium and phosphate decreases equally. For the thermal disequilibrium to enable sufficiently high phosphate concentrations for prebiotic chemistry (10–100 mM), acidic conditions must have existed beforehand for apatite dissolution....."

We thank the referee for the positive comment and for raising the requirement of a thorough discussion of the pH-dependent phosphate release.

As stated by the referee, the apatite solubility under acidic conditions is well known. In our work, we now address the case the case where the pH of this acidic, phosphate-enriched solution is increased again, e.g. by surface inflows, which is a strict prerequisite for many important synthesis pathways in prebiotic chemistry that would not work under acidic conditions.

Without the thermal non-equilibrium processes discussed in our work, both ions would re-precipitate and thus no longer be available for prebiotic chemistry (see also Fig. 2d, grey bars).

As novelty in our study, we show that the acidic-dissolved phosphate-calcium solution can be separated by simple thermal non-equilibria (Fig. 2c), ubiquitous in a variety of

geological settings. This previously unexplored separation of phosphate and calcium prior to the increase in pH (for usage of the phosphate by prebiotic reactions) leads to phosphate concentrations of up to 10 mM in the final, pH-neutral solution (Fig. 2d, red columns). This strongly deviates from the known equilibrium case where no phosphate remains in the pH-neutral solution.

These findings provide an unexpected heat-flow-driven key to the phosphate problem on the Early Earth, enabling the permanent solubilization of apatite as phosphate source for prebiotic chemistry at neutral pH.

To make this clearer in the discussion in the manuscript, we have modified the text:

“However, once the acidic-dissolved apatite solution was exposed to heat fluxes before reaching pH-neutral conditions, the selective thermophoretic removal of the co-dissolved calcium led to a 100-fold increase in phosphate solubility at neutral pH. We demonstrated that the achieved phosphate concentrations of up to 15 mM strongly enhanced the formation of TMP, an important activating and phosphorylation agent for driving various prebiotic reactions.”

In the introduction, we write:

“While apatite dissolves in acidic flows, the resulting solution would re-precipitate in shallow, neutral waters under thermal equilibrium conditions. We show that heat-flow-driven enrichment in acidic solutions can supply up to 15 millimolar of solubilized phosphate after close-to-surface pH neutralization [...]”

All in all, in my opinion, most of the concerns have been successfully addressed by the authors, although I still question the novelty of the work due to above mentioned reason and do not consider this as such a novel idea.

We hope that the above explanations and additions to the manuscript have now cleared up these unclear points.